# New Coelomycetous Fungi from Freshwater in Spain

**DOI:** 10.3390/jof7050368

**Published:** 2021-05-08

**Authors:** Viridiana Magaña-Dueñas, Alberto Miguel Stchigel, José Francisco Cano-Lira

**Affiliations:** Mycology Unit, Medical School and IISPV, Universitat Rovira i Virgili, Sant Llorenç 21, 43201 Reus, Tarragona, Spain; qfbviry@hotmail.com (V.M.-D.); jose.cano@urv.cat (J.F.C.-L.)

**Keywords:** Ascomycota, coelomycetous, freshwater fungi, phylogeny, plant debris, taxonomy

## Abstract

Coelomycetous fungi are ubiquitous in soil, sewage, and sea- and freshwater environments. However, freshwater coelomycetous fungi have been very rarely reported in the literature. Knowledge of coelomycetous fungi in freshwater habitats in Spain is poor. The incubation of plant debris, from freshwater in various places in Spain into wet chambers, allowed us to detect and isolate in pure culture several pycnidia-producing fungi. Fungal strains were phenotypically characterized, and a phylogenetic study was carried out based on the analysis of concatenated nucleotide sequences of the D1–D2 domains of the 28S nrRNA gene (LSU), the internal transcribed spacer region (ITS) of the nrDNA, and fragments of the RNA polymerase II subunit 2 (*rpb*2) and beta tubulin (*tub*2) genes. As a result of these, we report the finding of two novel species of *Neocucurbitaria*, three of *Neopyrenochaeta,* and one of *Pyrenochaetopsis*. Based on the phylogenetic study, we also transferred *Neocucurbitaria prunicola* to the genus *Allocucurbitaria*. This work makes an important contribution to the knowledge of the mycobiota of plant debris in freshwater habitats.

## 1. Introduction

Coelomycetous fungi are characterized by the production of conidia within a cavity lined by fungal or fungal-host tissue called conidiomata [1]. Conidiomata can be acervular (open, cup-shaped asexual fruiting bodies developing below the epidermis of the plant host tissue and bearing a series of adpressed conidiophores), pycnidial (globose, pyriform to flask-shaped asexual reproductive structures whose conidia are liberated through an usually apical opening [ostiolum]), or stromatic (consisting of undifferentiated sclerotic tissues, ostiolate or not, in which one or more lysigenic cavities develops, upholstered inside by conidiophores/conidiogenous cells forming conidia). Coelomycetous fungi are mostly parasites of terrestrial vascular plants but are also saprobic, growing at the expense of dead organic matter on the ground, especially on plant debris. These are ubiquitous on soil, sewage, and in salt- and freshwater environments. [2]. Freshwater coelomycetous fungi occur on stream-side plants or on submerged wood litter, and their conidia can also be recovered from foam and water samples [3]. Usually, they produce brown to blackish pycnidial fruiting bodies on submerged woody debris and stems of herbaceous plants, and produce several conidia from the conidiogenous cells [4]. Identification of coelomycetous fungi has gone through dramatic changes over the last decade, and currently involves DNA sequencing of several (four to six) genetic markers and the building of phylogenetic trees [5]. In Spain, there have been a few reports of coelomycetous fungi recovered from freshwater habitats. In 1990, Roldán and Honrubia reported *Bartalinia robillardoides* and *Truncatella angustata* [6], and Giralt described *Diplolaviopsis ranula* [7]. Up to 2014, only 16 coelomycetous fungi had been reported from freshwater habitats [4,8,9,10,11,12,13,14,15,16].

The main objective of this work was to characterize phenotypically and to identify molecularly those coelomycetous fungi found in different freshwater habitats in Spain.

## 2. Materials and Methods

### 2.1. Sampling and Fungal Isolation

A hundred samples of decomposing plant material submerged in freshwater habitats in Spain were collected: 3 from “Les Guilleries” (Barcelona province), 50 from “Cascadas del Huéznar” (Cazalla de la Sierra, Sevilla province, Spain), 17 from Riaza (Segovia province, Spain), and 30 from “Serra del Montsant” (Tarragona province, Spain). The samples were placed into self-sealing sterile plastic bags, which were closed and transported to the laboratory, and stored at room temperature (20−25 °C) until they were processed. The specimens were rinsed twice with 500 mL tap water, placed into Petri dishes or appropriate plastic containers lined inside with two sheets of filter paper, and moistened with sterile water with diehldrin^®^ (1 mL of a solution of 20 mg diehldrin^®^ in 20 mL of dimethyl-ketone/L of water), incubated at room temperature, and examined periodically under stereomicroscope for up to 2 months. Several propagules and/or fruiting bodies were taken and transferred using sterile disposable tuberculin-type needles to 55 mm diameter Petri dishes containing oatmeal agar (OA; 30 g of filtered oat flakes, 15 g agar-agar, 1 L tap water; [17]), and then incubated at room temperature. Once obtaining an axenic culture of each isolate, these were stored in the culture collection of the Faculty of Medicine of University Rovira i Virgili (FMR; Reus, Spain). Type specimens and ex-type cultures of the novel fungi were deposited in the Westerdijk Fungal Biodiversity Institute (CBS), Utrecht, The Netherlands (Appendix A).

### 2.2. Phenotypic Study

Macroscopic characterization of the colonies was performed on OA and on malt extract agar (MEA; Difco, Detroit, MI, USA) incubated for 14 d in the dark at 25 ± 1 °C [17]. Colony colour was determined according to Kornerup and Wanscher [18]. The ability of the isolates to grow at cardinal temperatures was determined on potato dextrose agar (PDA; Pronadisa, Madrid, Spain) after 7 d in the dark, ranging from 5 to 35 ± 1 °C at 5 °C intervals, plus 37 ± 1 °C [19]. Morphological characterization of vegetative and reproductive structures was performed growing the fungal strains on OA in the same conditions as for colony characterization, and examining at least 30 individuals of each structure [20,21] on Shear’s mounting medium (3 g potassium acetate, 60 mL glycerol, 90 mL ethanol 95%, and 150 mL distilled water; [22]) using a Olympus BH-2 bright field microscope (Olympus Corporation, Tokyo, Japan). Photomicrographs were taken using a Zeiss Axio-Imager M1 microscope (Oberkochen, Germany) with a DeltaPix Infinity X digital camera using Nomarski differential interference contrast.

### 2.3. DNA Extraction, Amplification and Sequencing

Fungal strains were cultured on PDA for 7 days at 25 ± 1 °C in the dark. Total DNA was extracted using the FastDNA kit protocol (Bio101, Vista, CA, USA) with a FastPrep FP120 instrument (Thermo Savant, Holbrook, NY, USA) according to the manufacturer’s protocol. DNA was quantified by using Nanodrop 2000 (Thermo Scientific, Madrid, Spain). The following *loci* were amplified and sequenced: LSU, with the primer pair LR0R [23] and LR5 [24]; ITS, with the primer pair ITS5 and ITS4 [25]; a fragment of the beta-tubulin gene (*tub*2) with the primers TUB2Fw and TUB4Rd [26]; and a fragment of the RNA polymerase II subunit 2 gene (*rpb*2) with RPB2-5F2 [27] and fRPB2-7cR primers [28]. The PCR amplifications were performed in a total volume of 25 µL containing 5 µL 10× PCR Buffer (Invitrogen, CA, USA), 0.2 mM dNTPs, 0.5 µL of each primer, 1 U Taq DNA polymerase, and 1−10 ng genomic DNA. PCR conditions for LSU, ITS, and *tub*2 were set as follows: an initial denaturation at 95 °C for 5 min; followed by 35 cycles of denaturation, annealing, and extension; and a final extension step at 72 °C for 10 min. For the LSU and ITS amplification, the 35 cycles consisted of 45 s at 95 °C, 45 s at 53 °C, and 2 min at 72 °C; and for the *tub*2 region 30 s at 94 °C, 45 s at 56°C, and 1 min at 72 °C. The PCR program for *rpb*2 amplification consisted of 5 cycles of 45 s at 94 °C, 45 s at 60 °C, and 2 min at 72 °C; then 5 cycles with 58 °C annealing temperature; and 30 cycles with a 54 °C annealing temperature. PCR products were purified and stored at −20 °C until sequencing. The same pairs of primers were used to obtain the sequences at Macrogen Spain (Macrogen Inc., Madrid, Spain). The consensus sequences were obtained using the SeqMan software v. 7 (DNAStar Lasergene, Madison, WI, USA).

### 2.4. Phylogenetic Analysis

We made a preliminary molecular identification by comparing the LSU, ITS, *tub*2, and *rpb*2 sequences of our isolates with those of the National Center for Biotechnology Information (NCBI) using the Basic Local Alignment Search Tool (BLAST; https://blast.ncbi.nlm.nih.gov/Blast.cgi (accessed on 16 March 2021)). For *tub*2 sequences, a maximum level of identity (MLI) of <98% provides identification only at genus level, and a value >98% was considered to allow for species-level identification. Alignment for each locus was performed with the MEGA (Molecular Evolutionary Genetics Analysis) software v. 7.0. (Tamura et al. 2013), using the ClustalW algorithm [29] and refined with MUSCLE [30] or manually, if necessary, on the same platform. Individual and concatenated phylogenetic trees were built after a maximum likelihood (ML) analysis carried out using the RAxML v. 8.2.10 [31] software on the online Cipres Science gateway portal [32], and a Bayesian Inference (BI) analysis using MrBayes v. 3.2.6 [33]. For ML analyses, the best nucleotide substitution model was General Time Reversible with Gamma distribution. Support for internal branches was assessed by 1000 ML bootstrapped pseudoreplicates. For the BI phylogenetic analysis, the best nucleotide substitution model was determined using jModelTest [34]. For ITS we used the symmetrical model with gamma distribution (SYM + G), for LSU and *tub*2 the symmetrical model with proportion of invariable sites and gamma distribution (SYM + I + G), and for *rpb*2 the symmetrical model with gamma distribution (SYM + G). The parameter settings used were two simultaneous runs of 5 M generations and four Markov chain Monte Carlo (MCMC), sampled every 1000 generations. The 50% majority-rule consensus tree and posterior probability values (PP) were calculated after discarding the first 25% of the samples. *Pleospora herbarum* CBS 191.86 and *P. typhicola* CBS 132.69 served as outgroup taxa. Confident branch support is defined as Bayesian posterior probabilities (PP) >0.95 and maximum likelihood bootstrap support (BS) >70%. Sequences generated in this study were deposited in European Nucleotide Archive (ENA), the final matrix used for phylogenetic analyses in TreeBASE (http://purl.org/phylo/treebase/phylows/study/TB2:S28077 (accessed on 16 March 2021)) and the novel taxonomic descriptions and nomenclature in MycoBank (www.mycobank.org (accessed on 16 March 2021)).

## 3. Results

### 3.1. Blast Search

Blast search results are shown in Appendix A.

### 3.2. Phylogenetic Relationships among Freshvwater Fungi

The final concatenated dataset obtained with both ML and BI analyses contained 71 in-groups of strains with a total of 2252 characters including gaps (455 for ITS, 791 for LSU, 272 for *tub*2, and 734 for *rpb*2), of which 704 are parsimony informative (170 for ITS, 69 for LSU, 143 for *tub*2, and 322 for *rpb*2). The sequence datasets did not show conflict in the tree topologies for the 70% reciprocal bootstrap trees, which allowed us to combine the four genes for the multi-locus analysis. The ML analysis showed similar tree topology and was congruent with that obtained in the BI. For the BI multi-locus analysis, a total of 11,663 trees were sampled after removal of the burn-in and reaching a stop value of 0.01. The support values were slightly different with the two analysis methods. In the phylogenetic tree (Figure 1), our strains were spread into three well-supported main clades, representing the families *Cucurbitariaceae* (99% BS/1 PP), *Neopyrenochaetaceae* (98% BS/1 PP), and *Pyrenochaetopsisaceae* (100% BS/1 PP). The *Cucurbitariaceae* clade was divided into four well-supported clades corresponding to the accepted genera (*Neocucurbitaria*, 100% BS/1 PP; *Paracucurbitaria*, 100% BS/1 PP; *Cucurbitaria*, 100% BS/1 PP and *Allocucurbitaria*, 95% BS/1 PP). The *Neocucurbitaria* clade was represented by all previously described species and three of our strains, all placed in independent terminal branches. The clade corresponding to the genus *Allocucurbitaria* included the type species *A. botulispora* and the new combination *A. prunicola* (basionym *Neocucurbitaria prunicola*). The *Neopyrenochaetaceae* clade included 11 species of the genus *Neopyrenochaeta*. Five of our strains resulted as co-specific with *N. annellidica* and *N. maesuayensis*, whereas the other three strains were each placed into independent terminal branches. The family *Pyrenochaetopsisaceae* was divided in two clades, corresponding to the genera *Neopyrenochaetopsis* and *Pyrenochaetopsis* (100% BS/1PP). *Pyrenochaetopsis* encompassed 16 species and our strain FMR 17327, which is located in an independent branch. Single gene-based phylogenies are shown as Appendix A because they resulted in being less informative and resolutive than those based on the four-loci concatenated tree.

### 3.3. Taxonomy

**Dothideomycetes**.

***Cucurbitariaceae*** G. Winter (as *Cucurbitarieae*), Rabenh. Krypt.-Fl., Edn 2 (Leipzig) 1.2: 308 (1885). MycoBank MB 80667.

Type genus: *Cucurbitaria* Gray, Nat. Arr. Brit. Pl. (London) 1: 519 (1821).

***Neocucurbitaria*** Wanas., E.B.G. Jones & K.D. Hyde, in Wanasinghe, Phookamsak, Jeewon, Li, Hyde, Jones, Camporesi & Promputtha, Mycosphere 8(3): 408 (2017). MycoBank MB 552832.

Type species: *Neocucurbitaria unguis-hominis* (Punith. & M.P. English) Wanas., E.B.G. Jones & K.D. Hyde, in Wanasinghe, Phookamsak, Jeewon, Li, Hyde, Jones, Camporesi & Promputtha, Mycosphere 8(3): 412 (2017). MB 552835.

= *Pyrenochaeta unguis-hominis* Punith. & M.P. English, Transactions of the British Mycological Society 64 (3): 539 (1975). MB 322137.

***Neocucurbitaria variabilis*** V. Magaña-Dueñas, Stchigel & Cano, ***sp. nov*.** Figure 2. MycoBank MB 838833.

*Etymology*: From Latin *variabilis*, due to the variable shape of the conidiogenous cells.

*Type*: Spain, Segovia province, Riaza, from plant debris in freshwater, May 2018, Viridiana Magaña Dueñas, holotype CBS H-24739, culture ex-type FMR 17552.

*Description*: Hyphae hyaline to pale brown; septate; branched; smooth- and thin-walled; 2–5 µm wide; with short, finger-like lateral projections; anastomosing. Conidiomata pycnidial, brown to dark brown, immersed to semi-immersed, solitary, scattered, setose, ostiolate, subglobose to globose, 110−120 µm × 120–140 µm, ostiole 40−50 µm diameter. Setae pale brown, erect, septate, smooth- and thick-walled, rounded at the tip, 40−80 µm × 3−4 µm. Conidiomata wall composed of three to five layers of cells, 15–25 µm thick, covered by a mass of interwoven, pale brown to brown hyphae; outer layer of *textura angularis*, composed of brown to dark brown, flattened polygonal cells of 3.5–4.5 µm diameter. Conidiophores absent. Conidiogenous cells phialidic, determinate, hyaline and smooth-walled, flask-shaped, 5–6 µm × 2–3 µm, or elongate-cylindrical, straight, sinuous or slightly curved, 10−14 µm × 1.5−3 µm. Conidia one-celled, hyaline, smooth- and thin-walled, ellipsoidal, ovoid or kidney-shaped, 2.5–3.5 µm × 1.0–1.5 µm. Chlamydospores absent.

*Culture characteristics*: Colonies on PDA reaching 22 mm diameter after 7 days at 25 ± 1 °C, flattened, velvety, margin regular, surface and reverse yellowish grey (4B2). Colonies on OA reaching 20 mm diameter after 7 days at 25 ± 1 °C, flattened, floccose, margin regular, grey to brownish grey (4F1/4D2); reverse grey to yellowish grey (4F1/4B2). Colonies on MEA reaching 16 mm diameter after 7 days at 25 ± 1 °C, umbonate, velvety, margin regular, yellowish grey to olive brown (4B2/4D3); reverse brownish grey to yellowish grey (7F2/4B2). Exopigment absent. Cardinal temperatures of growth: minimum 5 °C, optimum 25 °C, and maximum 30 °C.

*Other material examined*: Spain, Sevilla province, Parque Natural Sierra Norte (37.994712, −5.668709), from plant debris in freshwater, May 2019, José F. Cano Lira, living cultures FMR 17877.

Diagnosis: In our phylogenetic tree, N. variabilis was placed in a terminal branch within the same subclade as *N. acerina*, *N. aquadulcis*, *N. aquatica*, *N. irregularis*, *N. keratinophila*, and *N. unguis-hominis*. *Neocucurbitaria variabilis* differs morphologically from all these species in having two kinds of enteroblastic conidiogenous cells, ampulliform and elongate-cylindrical, which are discrete; and doliiform in *N. acerina*, *N. aquatica* and *N. irregularis*, and integrated in acropleurogenous conidiophores (i.e., having terminal and lateral openings) in *N. keratinophila* and *N. unguis-hominis* [35].

*Notes*: Differences in nucleotide sequences (ITS-LSU-*tub*2-*rpb*2 concatenated dataset; 2252 bp) between *N. variabilis* and the species in the same terminal clade are: *N. aquadulcis*, 71 bp; *N. keratinophila*, 77 bp; *N. irregularis*, 78 bp; *N. acerin*a, 80 bp; *N*. *aquatica*, 82 bp; and *N. unguis-hominis*, 83 bp.

***Neocucurbitaria aquadulcis*** V. Magaña-Dueñas, Cano & Stchigel, *sp. nov*. Figure 3. MycoBank MB 838834.

*Etymology*: From Latin *aqua*-, water; and -*dulcis*, sweet, because of the origin of the fungus.

*Type*: Spain, Sevilla province, Parque Natural Sierra Norte (37.931670, −5.704493), from plant debris in freshwater, May 2019, José F. Cano Lira, holotype CBS H-24740, culture ex-type FMR 17840 = CBS 147605.

*Description*: Hyphae hyaline to light brown, septate, branched, smooth- and thin-walled, 2–3 µm wide, anastomosing. Conidiomata pycnidial, brown to dark brown, immersed to semi-immersed, solitary or confluent, scattered, ostiolate, covered by a mass of interwoven, pale brown to brown hyphae, ovoid to globose, 120–150 µm × 130–170 µm, ostiole of 15–20 µm diameter. Conidiomata wall composed of three to six layers of cells, 15–35 µm thick, outer layer of *textura angularis*, composed of brown to dark brown, flattened polygonal cells of 3–5.5 µm diameter. Conidiophores absent. Conidiogenous cells phialidic, determinate, hyaline, smooth-walled, ampulliform, 4–6 µm × 2–3 µm. Conidia one-celled, hyaline, smooth- and thin-walled, bacillary, slightly curved, 2.5–4.5 µm × 1.0–2.0 µm. Chlamydospores absent.

*Culture characteristics*: Colonies on PDA reaching 6–9 mm diameter after 7 days at 25 ± 1 °C, convex, granular, margins irregular, surface, and reverse olive brown (4D5). Colonies on OA reaching 10–11 mm diameter after 7 days at 25 ± 1 °C, flattened, velvety, margin regular, surface and reverse yellowish brown (5F5) to brownish grey (5B2). Colonies on MEA reaching 10–15 mm diameter after 7 days at 25 ± 1 °C, umbonate, velvety, margins regular, brownish grey to orange grey (5E2/5B2) reverse yellowish brown to orange grey (5E4/5B2). Exopigment not produced. Cardinal temperatures of growth: minimum 5 °C, optimum 25 °C, and maximum 30 °C.

*Diagnosis*: *Neocucurbitaria aquadulcis*, unlike *N. variabilis*, produces solely ampulliform phialides (see before) and bigger conidia (2.5–4.5 µm × 1.1–1.9 µm vs. 2.5–3.5 µm × 1.2–1.7 µm). Also, the conidiomata wall of *N. aquadulcis* is covered by hyphae, whereas it is setose in *N. variabilis*.

*Notes*: Differences in nucleotide sequences (ITS-LSU-*tub*2-*rpb*2 concatenated dataset; 2252 bp) between *N. aquadulcis* and the species in the same terminal clade are: *N. keratinophila*, 54 bp; *N. irregularis*, 57 bp; *N. acerina*, 61 bp; *N. aquatica* and *N. unguis-hominis*, 62 bp; and *N. variabilis*, 71 bp.

***Allocucurbitaria*** Valenz.-Lopez, Stchigel, Guarro & Cano, in Valenzuela-Lopez, Cano-Lira, Guarro, Sutton, Wiederhold, Crous & Stchigel, Stud. Mycol. 90:51 (2017). MycoBank MB 821455.

*Type species*: *Allocucurbitaria botulispora* Valenz.-Lopez, Stchigel, Guarro & Cano, in Valenzuela-Lopez, Cano-Lira, Guarro, Sutton, Wiederhold, Crous & Stchigel, Stud. Mycol. 90:51 (2017). MycoBank MB 819770.

***Allocucurbitaria prunicola*** (Crous & Akulov) V. Magaña-Dueñas, Stchigel & Cano, *comb. nov.* MycoBank MB 838843.

*Basionym*: *Neocucurbitaria prunicola* Crous & Akulov, in Crous, Schumacher, Akulov, Thangavel, Hernández-Restrepo, Carnegie, Cheewangkoon, R; Wingfield, Summerell, Quaedvlieg, Coutinho, Roux, Wood, Giraldo & Groenewald, Fungal Systematics and Evolution 3:91 (2019).

*Description*: Crous & Akulov 2019.

*Notes*: In 2019, Crous & Akulov introduced *N. prunicola* to the genus *Neocucurbitaria*, based on morphological and nucleotide sequence data analysis [36]. However, in our phylogenetic study, *N. prunicola* is clearly placed in the genus *Allocucurbitaria*. Therefore, we propose a new combination for that species.

***Neopyrenochaetaceae*** Valenz.-Lopez, Crous, Stchigel, Guarro & Cano, in Valenzuela-Lopez, Cano-Lira, Guarro, Sutton, Wiederhold, Crous & Stchigel, Stud. Mycol. 90:54 (2017). MycoBank MB 820416.

*Type genus*: *Neopyrenochaeta* Valenz.-Lopez, Crous, Stchigel, Guarro & Cano, in Valenzuela-Lopez, Cano-Lira, Guarro, Sutton, Wiederhold, Crous & Stchigel, Stud. Mycol. 90:54 (2017). MycoBank MB 820313.

*Type species*: *Neopyrenochaeta acicola* (Moug. & Lév.) Valenz.-Lopez, Crous, Stchigel, Guarro & Cano, in Valenzuela-Lopez, Cano-Lira, Guarro, Sutton, Wiederhold, Crous & Stchigel, Stud. Mycol. 90:54 (2017). MycoBank MB 820314.

***Neopyrenochaeta asexualis*** V. Magaña-Dueñas, Stchigel & Cano, *sp. nov.* Figure 4. MycoBank MB 838835.

*Etymology*: From Latin *asexualis*, without sex, because of lack of a known sexual morph.

*Type*: Spain, Sevilla province, Parque Natural Sierra Norte (37.994712, −5.668709), from plant debris in freshwater, May 2019, José F. Cano Lira, holotype CBS H-24741, culture ex-type FMR 17874 = CBS 147606.

*Description*: Hyphae hyaline to light brown, septate, branched, smooth-and thin-walled, 2–3 µm wide. Conidiomata pycnidial, brown to dark brown, immersed to semi-immersed, solitary or confluent, setose, ostiolate, globose to subglobose, 100–170 µm × 85–150 µm, ostiole 30–40 µm diameter. Setae pale brown to brown, septate, erect, nodose, narrowing towards the tip, thick-walled 40–80 µm × 2–4 µm, mainly disposed around the ostiole but also scattered, sometimes curved or recurved at the tip. Conidiomata wall composed of two to four layers of cells, 10–25 µm thick, outer layer of *textura angularis*, composed of brown to dark brown, flattened polygonal cells of 5–7 µm diameter. Conidiophores absent. Conidiogenous cells phialidic; determinate; hyaline; smooth-walled; doliiform; 5–6 µm × 4–5 µm; with mostly one, less frequently two conidiogenous *loci*. Conidia one-celled, hyaline, smooth- and thin-walled, ellipsoidal, 4–5 µm × 1.5–2.5 µm, sometimes slightly curved. Chlamydospores absent.

*Culture characteristics*: Colonies on PDA reaching 36 mm diameter after 7 days at 25 ± 1 °C, umbonate, velvety, margins regular, brownish grey (4D2), with patches of white; reverse olive brown (4D3). Colonies on OA reaching 40 mm diameter after 7 days at 25 ± 1 °C, flattened to slightly floccose, margin regular, with sparse aerial mycelium, grey (30C1); reverse greenish grey to yellowish grey (30E1/30F2). Colonies on MEA reaching 25 mm diameter after 7 days at 25 ± 1 °C, convex, velvety, margin regular, golden grey (4C2); reverse brownish grey to beige (4F2/4C3). Exopigment absent. Cardinal temperatures of growth: minimum 5 °C, optimum 25 °C, and maximum 30 °C.

*Diagnosis*: *Neopyrenochaeta asexualis* is grouped in the same terminal clade as *N. thailandica*, but as a distinct taxon. Morphological comparison between *N. asexualis* and *N. thailandica* is not possible because only the former produces the asexual morph and only the latter one forms ascomata [37]. However, it is noteworthy that *N. asexualis* produces conidiomata with doliiform phialides with up two conidiogenous *loci*.

*Notes*: The difference in nucleotide sequences (ITS-LSU-*tub*2-*rpb*2 concatenated dataset) between *N. asexualis* and *N. thailandica* is 38 bp.

***Neopyrenochaeta submersa*** V. Magaña-Dueñas, Cano & Stchigel, *sp. nov.* Figure 5. MycoBank MB 838840.

*Etymology*: From Latin *submersum*, submerged, because the fungus was recovered from plant debris in freshwater.

*Type*: Spain, Barcelona province, Les Guilleries (41.9362028, 2.4122862), from plant debris in freshwater, Nov 2017, Eduardo Jose de Carvalho Reis, holotype CBS H-24742, culture ex-type FMR 16957 = CBS 147607.

*Description*: Hyphae pale to dark brown, septate, branched, smooth- and thin-walled, 2–3 µm wide. Conidiomata pycnidial, brown to dark brown, semi-immersed, solitary or confluent, scattered, ostiolate, setose, globose to subglobose, 140–200 µm × 180–240 µm, one to three ostioles per conidioma, 60–85 µm diameter. Setae brown to dark brown, septate, erect, rounded at the tip, thick-walled, 75−160 µm × 2−3 µm, narrowing towards the tip, and mostly disposed around the ostiole. Conidiomata wall composed of three to five layers of cells, 10–20 µm thick, with an outer layer of *textura angularis*, composed of brown to dark brown, flattened polygonal cells of 3–4 µm diameter. Conidiophores absent. Conidiogenous cells phialidic, determinate, hyaline, smooth-walled, ampulliform, 6–8 µm × 2.5–3.5 µm. Conidia one-celled, hyaline, smooth- and thin-walled, ellipsoidal, 3–4 µm × 2–3 µm. Chlamydospores absent.

*Culture characteristics*: Colonies on PDA reaching 23 mm diameter after 7 days at 25 ± 1 °C, umbonate, velvety, rugose, margin regular, surface greyish green to greenish grey (30E4/30C2), reverse greyish green to dull green (30B4/30E3), margin greenish grey (30C2). Colonies on OA reaching 28 mm diameter after 7 days at 25 ± 1 °C, convex, velvety, margin regular, surface and reverse grey (30C1). Colonies on MEA reaching 20 mm diameter after 7 days at 25 ± 1 °C, umbonate, velvety, margins regular, greyish green to dull green (30C3/30E3), margin white; reverse dark green (30F3), margins white. Exopigment absent. Cardinal temperatures of growth: minimum 5 °C, optimum 25 °C, and maximum 30 °C.

*Diagnosis*: In our phylogenetic analysis, *N. submersa* is located in the same terminal clade as *N. acicola*, *N. fragariae*, *N. inflorescentiae*, and *N. glabra*. With the exception of *N. glabra*, all these species are morphologically very similar. However, *N. submersa* grows faster than *N. fragariae* (reaching 14 mm and 11 mm diameter after 7 days at 25 °C on OA and MEA, respectively). Also, *N. submersa* does not produce exopigment on OA, which is lilac-rose in *N. acicola* [38] and orange in *N. fragariae*.

*Notes*: Differences in nucleotide sequences (ITS-LSU-*tub*2-*rpb*2 concatenated dataset) between *N. submersa* and the other species in the same terminal clade are: *N. fragariae* 28 bp; *N. acicula*, 33 pb; *N. inflorescentiae*, 51 pb; and *N. glabra*, 98 bp.

***Neopyrenochaeta glabra*** V. Magaña-Dueñas, Stchigel & Cano, *sp. nov.* Figure 6. MycoBank MB 838841.

*Etymology*: From Latin *glaber*, hairless, relating to absence of setae.

*Type*: Spain, Segovia province, Riaza (41.238863, -3.435258), from freshwater submerged plant debris, May 2018, Viridiana Magaña Dueñas, holotype CBS H-24743, culture ex-type FMR 17418 = CBS 147608.

*Description*: Hyphae hyaline to pale brown, septate, branched, smooth- and thin-walled, 1.5–2.5 µm wide. Conidiomata pycnidial immersed to semi-immersed, solitary or confluent, scattered, ostiolate, glabrous, translucent, pale brown to brown, but carbonaceous around the ostiole, mostly subglobose, 140–200 µm × 180–240 µm, ostiole 60–85 µm diam. Conidiomata wall composed of four to six layers of cells, 10–25 µm thick, with an outer layer of *textura angularis*, composed of pale brown to brown, flattened polygonal cells of 3–4 µm diameter. Conidiophores absent. Conidiogenous cells phialidic, determinate, hyaline, smooth-walled, ampulliform, 7–9 µm × 3–4 µm. Conidia aseptate, hyaline, smooth- and thin-walled, ellipsoidal, 3–4 µm × 2–3 µm. Chlamydospores absent.

*Culture characteristics*: Colonies on PDA reaching 23 mm diameter after 7 days at 25 ± 1 °C, umbonate, velvety, rugose, margin regular, surface, and reverse grey to dark brown (8F1/8D1). Colonies on OA reaching 25 mm diameter after 7 days at 25 ± 1 °C, flattened, velvety, margin regular, surface and reverse grey (6F1). Colonies on MEA reaching 23 mm diam after 7 days at 25 ± 1 °C, convex, velvety, margins regular, grey to olive brown (4F1/4D3), margin yellowish grey (4B2); reverse brownish grey (5E2) margins orange grey (5B2). Exopigment absent. Cardinal temperatures of growth: optimum 25 °C, maximum 30 °C, minimum 5 °C.

*Diagnosis*: Morphologically, *N. glabra* differs from the phylogenetically closest species, *N. acicula*, *N. fragariae, N. inflorescentiae,* and *N. submersa* by lacking conidiomatous setae in the conidiomata walls.

*Notes*: Differences in nucleotide sequences (ITS-LSU-*tub*2-*rpb*2 concatenated dataset) between *N. glabra* and the other species of the same terminal clade are: *N. fragariae*, 89 bp; *N. submersa*, 98 bp; *N. acicola*, 108 bp; and *N. inflorescentiae*, 131 bp.

***Pyrenochaetopsidaceae*** Valenz.-Lopez, Crous, Cano, Guarro & Stchigel, in Valenzuela-Lopez, Cano-Lira, Guarro, Sutton, Wiederhold, Crous & Stchigel, Stud. Mycol. 90:56 (2017). MycoBank MB 820308.

Type genus: *Pyrenochaetopsis* Gruyter, Aveskamp & Verkley, Mycologia 102 (5):1076 (2010). MycoBank MB 514653.

Type species: *Pyrenochaetopsis lep.tospora* (Sacc. & Briard) Gruyter, Aveskamp & Verkley, Mycologia 102 (5):1076 (2010). MycoBank MB 514654.

***Pyrenochaetopsis aquatica*** V. Magaña-Dueñas, Cano & Stchigel, *sp. nov*. Figure 7. MycoBank MB 838842.

*Etymology*: From Latin *aquaticus*, referring to the habitat from which the fungus was recovered (freshwater).

*Type*: Spain, Tarragona province, Serra del Montsant (41.32871, 0.87105), from plant debris in freshwater, February 2018, Eduardo Jose de Carvalho Reis, holotype CBS H-24744, culture ex-type FMR 17327 = CBS 147609.

*Description*: Hyphae hyaline to pale brown, septate, branched, smooth- and thin-walled, 1.5–2 µm wide. Conidiomata pycnidial, brown, immersed to semi-immersed, solitary or confluent, scattered, ostiolate, mostly glabrous or covered with few short setae, pyriform, 200–300 µm × 130–180 µm, ostiole 60–80 µm diameter. Setae pale brown to brown, septate, erect, nodose, thick-walled, of 10–20 µm × 3–4 µm, tapering towards the apex, mainly disposed around the ostiole. Conidiomata wall composed of four to six layers of cells, 15–30 µm thick, with an outer layer of *textura angularis*, composed of brown to dark brown, flattened polygonal cells of 3.5–4.5 µm diameter. Conidiophores absent. Conidiogenous cells phialidic, determinate, hyaline, smooth-walled, ampulliform, 6–7 µm × 2.5–3 µm. Conidia aseptate, hyaline, smooth- and thin-walled, mostly long ellipsoidal, 3.5–5 µm × 1–1.8 µm, slightly constricted at the middle, sometimes slightly curved and irregularly shaped, biguttulate. Chlamydospores absent.

*Culture characteristics*: Colonies on PDA reaching 15 mm diameter after 7 days at 25 ± 1 °C, flattened, velvety, margin regular, grey to yellowish white (A1C/4A2); reverse greyish green to yellowish white (4B3/4A2). Colonies on OA reaching 20 mm diameter after 7 days at 25 ± 1 °C, flattened, velvety, margin regular, yellowish grey to yellowish white (4B2/4A2). Colonies on MEA reaching 20 mm diameter after 7 days at 25 ± 1 °C, umbonate, velvety, margins regular, yellowish grey to yellowish brown (4B2/5F4), margin orange grey (5B2); reverse orange grey (5B2). Exopigment absent. Cardinal temperatures of growth: optimum 25 °C, maximum 30 °C, minimum 5 °C.

*Diagnosis*: *Pyrenochaetopsis aquatica* differs morphologically from the phylogenetically nearest species *P. leptospora* and *P. poae*, because it is mostly glabrous or covered with few short setae, while the pycnidia of *P. leptospora* and *P. poae* are abundantly covered with long setae [39].

*Notes*: Differences in nucleotide sequences (ITS-LSU-*tub*2-*rpb*2 concatenated dataset) between *P. aquatica* and the other species of the same terminal clade are: *P. leptospora*, 57 bp; and *P. poae*, 67 bp.

## 4. Discussion

The genus *Neocucurbitaria* was introduced by Wanasinghe et al. [40] to accommodate *N. acerina*, *N. quercina* and *N. unguis-hominis* (the type species of the genus). Twenty-two species are currently accepted (Index of Fungi; http://www.indexfungorum.org/names/Names.asp (accessed on 16 March 2021)). *Neocucurbitaria* spp. has been isolated from human corneal and skin lesions, seawater, and trees and shrubs [5,40,41]. We described two new species for the genus, *N. aquadulcis* and *N. variabilis*, from submerged plant debris in freshwaters, the first report for this sort of habitat. It is remarkable that *N. variabilis* produces two sorts of conidiogenous cells (flask-shaped and long cylindrical) and that *N. aquadulcis* only produces ampulliform phialides, whereas the other species in the same subclade (*N. acerina*, *N. aquatica*, *N. irregularis*, *N. keratinophila* and *N. unguis-hominis*) produce doliiform phialides or well-developed conidiophores. In 2019, Crous & Akulov introduced *N. prunicola* to that genus [36]. However, in our phylogenic analysis, *N. prunicola* was located far from the type species of *Neocucurbitaria* (*N. unguis-hominis*), being located within the genus *Allocucurbitaria*. Consequently, we propose the new combination *Allocucurbitaria prunicola*.

A molecular study by Valenzuela-López et al. [5] allowed recognition of four new families of coelomycetous fungi included previously in the family *Cucurbitariaceae*: *Neopyrenochaetaceae*, *Parapyrenochaetaceae*, *Pseudopyrenochaetaceae,* and *Pyrenochaetopsidaceae*. In the latter family, the authors recognized four species belonging to the genus *Neopyrenochaeta*: *N. acicola* (basionym: *Vermicularia acicola*; originally described on decaying leaves of *Pinus sylvestris*, Vosges, France), *N. fragariae* (originally identified as *Pyrenochaeta acicola*; isolated from *Fragaria* (×) *ananassa*, The Netherlands), *N. inflorescentiae* (basionym: *Pyrenochaeta inflorescentiae*; from style of senescent flowerhead of *Protea neriifolia*, Western Cape Province, South Africa), and *N. thelephonii* (basionym: *Pyrenochaeta telephonii*; from surface of cell phone, Maharashtra, India) [40,42,43]. During 2019 and 2020, eight more species were described [37,44], three of them (*Neopyrenochaeta annellidica*, *Neopyrenochaeta chiangraiensis* and *Neopyrenochaeta maesuayensis*) from submerged decaying wood in Thailand. Interestingly, we also identified two of these latter three species in Spain (Figure 1). This implies that the geographical distribution of *N. annellidica* and *N. maesuayensis* is much broader than would be expected, since their original report was from tropical areas of Southeastern Asia. In the present study, we report the finding of three novel additional species from submerged plant debris in Spain: *Neopyrenochaeta glabra*, *N. asexualis,* and *N. submersa*. *Neopyrenochaeta glabra* is easily recognized by the absence of setae and the darker conidiomata wall around the ostiole. *Neopyrenochaeta asexualis* is distinguished from other species of the genus because it produces doliiform phialides with one or two conidiogenous loci. Otherwise, *N. submersa* is difficult to discriminate morphologically from *N. acicola*, *N. fragariae,* and *N. inflorescentiae* species phylogenetically related but differing molecularly.

The fungal genus *Pyrenochaetopsis* was introduced by De Gruyter et al. [45] to accommodate: *P. decipens, P. indica, P. leptospora* (type species of the genus), *P. microspore,* and *P. pratorum*. Currently 16 species are accepted (Index Fungorum 2020). The members of this genus have been found in terrestrial and marine environments, human dermatitis, sputum, and blood human samples, [5,37,46,47,48,49]. In our phylogenetic analysis, the strain FMR 17337, named here as *P. aquatica*, clustered within the *Pyrenochaetopsis* clade, is distant from other species of this genus, with the exception of *P. leptospora* and *P. poae*, which forms a sister clade. Both species differ phylogenetically and morphologically from *P. aquatica* in having more abundant and longer setae.

## 5. Conclusions

In the present study, we have isolated several coelomycetous fungi from submerged plant debris collected in different freshwater habitats in Spain by incubation of the samples in wet chambers. After a phenotypic characterization and a phylogenic study based on the analysis of nucleotide sequences of the ITS, LSU, *tub*2, and *rpb*2 loci, six new species have been described: *Neocucurbitaria aquadulcis* and *N. variabilis*; *Neopyrenochaeta glabra*, *N. asexualis* and *N. submersa*; and *Pyrenochaetopsis aquatica*. Also, thanks to the phylogenetic analysis, *Neocucurbitaria prunicola* was transferred to the genus *Allocucurbitaria*. In our opinion, the present study makes an important contribution to the knowledge of the coelomycetous fungi growing on decomposing plant material in aquatic habitats.

## Figures and Tables

**Figure 1 jof-07-00368-f001:**
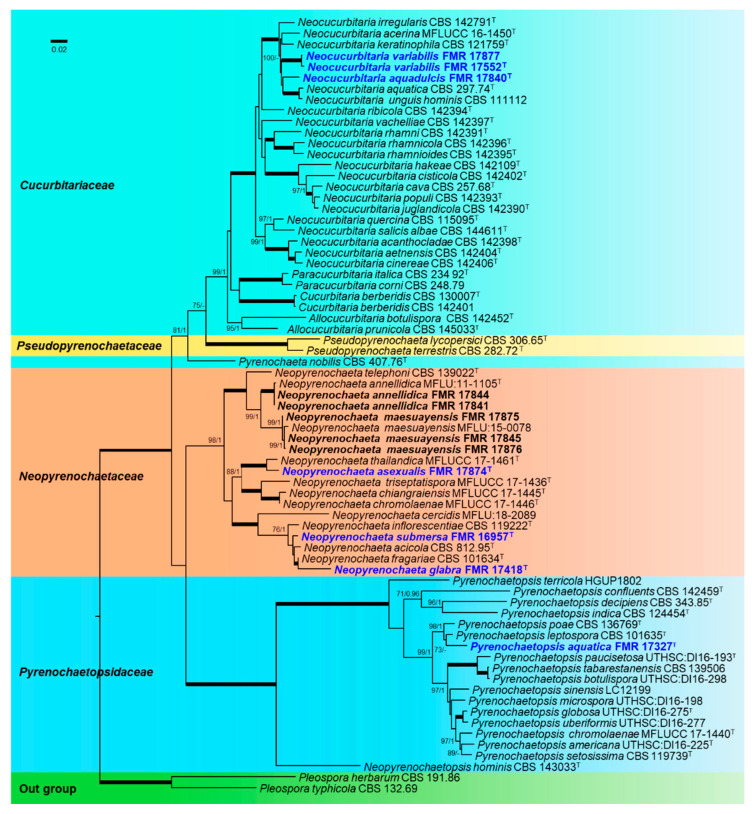
ML phylogenetic tree of *Cucurbitariaceae*, *Pseudopyrenochaetaceae*, *Neopyrenochaetaceae*, and *Pyrenochaetopsidaceae* inferred from the combined sequences of ITS, LSU, *tub*2, and *rpb*2 loci. Support in nodes is indicated above branches and is represented by posterior probabilities (BI analysis) of 0.95 and higher, and/or bootstrap values (ML analysis) of 70% and higher. Full-supported branches (100% BS/1 PP) are indicated by **thicker lines**. ^T^ =ex-type strains. New species are indicated in **blue**. New strains isolated during this study are indicated in **bold**. Alignment length 2252 bp. The sequences not generated by us were retrieved from EMBL/GenBank and are indicated in Appendix A.

**Figure 2 jof-07-00368-f002:**
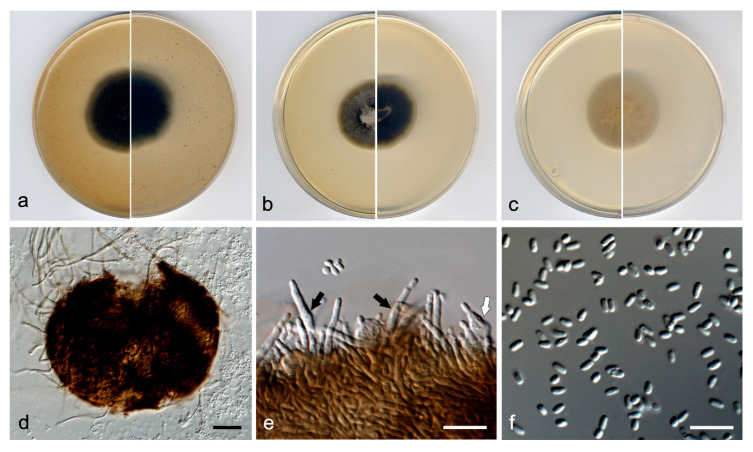
*Neocucurbitaria variabilis* FMR 17552 ^T^. (**a**) Colonies on OA, (**b**) MEA, and (**c**) PDA, after 2 weeks at 25 ± 1 °C (surface, left; reverse, right); (**d**) pycnidium; (**e**) conidiogenous cells (black arrow, elongated cylindrical; white arrow, flask-shaped); (**f**) conidia. Scale bars: d = 50 µm, e,f = 10 µm.

**Figure 3 jof-07-00368-f003:**
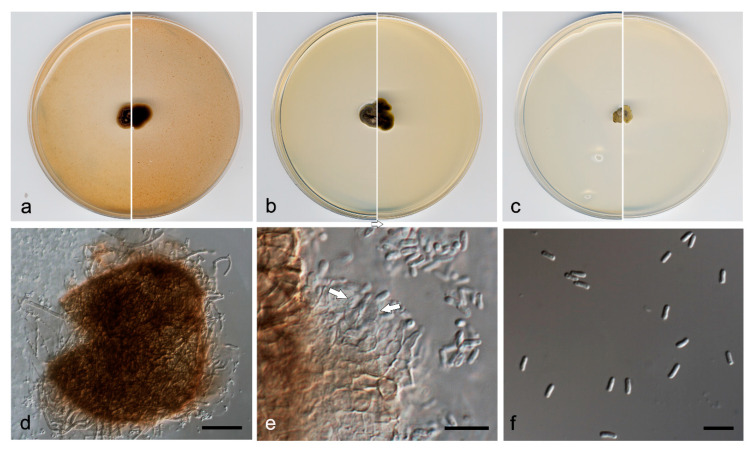
*Neocucurbitaria aquadulcis* FMR 17840 ^T^. (**a**) Colonies on OA, (**b**) MEA, and (**c**) PDA, after 2 weeks at 25 ± 1 °C (surface, left; reverse, right); (**d**) pycnidium; (**e**) conidiogenous cells (white arrows); (**f**) conidia. Scale bars: d = 50 µm, e,f = 10 µm.

**Figure 4 jof-07-00368-f004:**
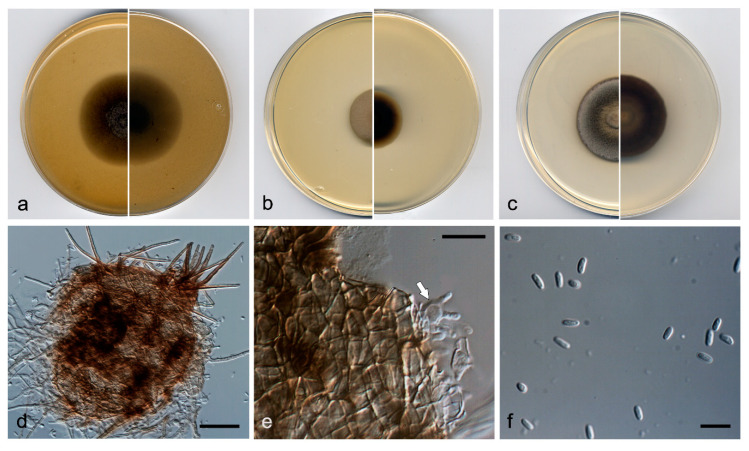
*Neopyrenochaeta asexualis* FMR 17874 ^T^. (**a**) Colonies on OA, (**b**) MEA, and (**c**) PDA, after 2 weeks at 25 ± 1 °C (surface, left; reverse, right); (**d**) pycnidium; (**e**) conidiogenous cells (white arrow); (**f**) conidia. Scale bars: d = 50 µm, e,f = 10 µm.

**Figure 5 jof-07-00368-f005:**
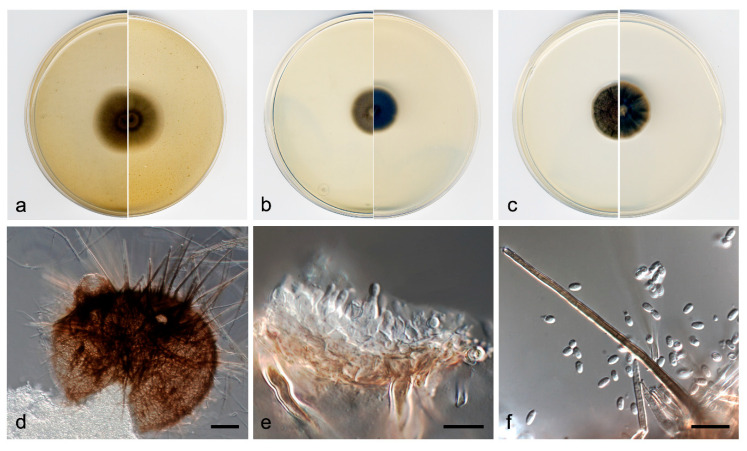
*Neopyrenochaeta submersa* FMR 16957 ^T^. (**a**) Colonies on OA, (**b**) MEA, and (**c**) PDA, after 2 weeks at 25 ± 1 °C (surface, left; reverse, right); (**d**) pycnidium; (**e**) conidiogenous cells; (**f**) conidia and setae. Scale bars: d = 50 µm, e,f = 10 µm.

**Figure 6 jof-07-00368-f006:**
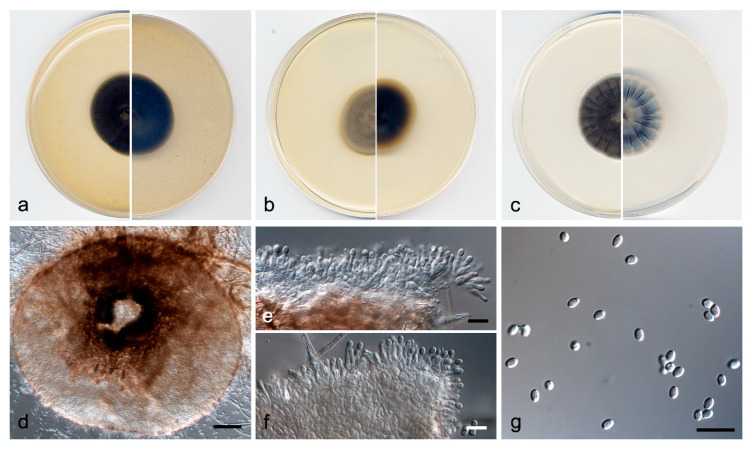
*Neopyrenochaeta glabra* FMR 17418 ^T^. (**a**) Colonies on OA, (**b**) MEA, and (**c**) PDA, after two weeks at 25 ± 1 °C (surface, left; reverse, right); (**d**) pycnidium; (**e**,**f**) conidiogenous cells; (**g**) conidia. Scale bars: d = 50 µm, e–g = 10 µm.

**Figure 7 jof-07-00368-f007:**
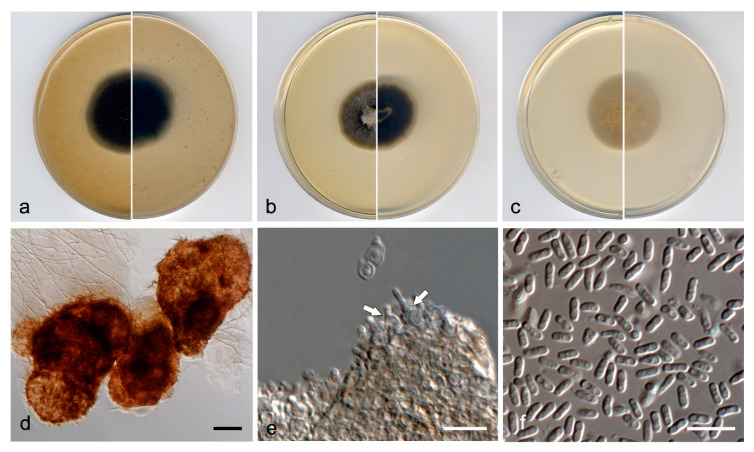
*Pyrenochaetopsis aquatica* FMR 17327 ^T^. (**a**) Colonies on OA, (**b**) MEA, and (**c**) PDA, after 2 weeks at 25 ± 1 °C (surface, left; reverse, right); (**d**) pycnidium; (**e,f**) conidiogenous cells (white arrows).

## Data Availability

Not applicable.

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
