# Peer review of "New Coelomycetous Fungi from Freshwater in Spain"

_jof, 2021, doi:10.3390/jof7050368_

Round 1

Reviewer 1 Report

COMMENTS TO THE MANUSCRIPT “New coelomycetous fungi from freshwater in Spain” by Magaña-Dueñas et al.

General comment:

In the submitted manuscript the authors isolate and analyze fungal Coleomycetes from freshwater ecosystems from Spain. Coleomycete taxa are scarcely studied worldwide, and poorly described in Spain. In the submitted work, an axenic mycelial culture of several Coleomycete fungi were isolated from plant debris submerged in freshwater. A detailed and well-performed mycelial colony morphological and molecular phylogenetic analysis were conducted on the obtained isolates. Based on the obtained results, six new species distributed in three different genera and three different families were proposed. A new combination of one species was also proposed. Additional two Neopyrenochaeta species were described for first time to Spain.  The submitted manuscript is well written, the methodology is properly described, and the results clearly explained and discussed.

The description of new species or knowing ones along different ecosystems and in new geographical areas is of scientific concern to continuously increase the knowledge on the diversity of this relatively unknow fungal taxon. By this, the subject of the submitted manuscript possess fungal taxonomy and diversity interest and is suitable to be published in the Journal of Fungi. However, the manuscript only needs minor review and optimization before being accepted.

Specific comments:

1. Recent papers have emphasized the relevance to include georeferencing metadata associated to fungal taxonomic and biodiversity studies, georeferenced fungal specimens (sporocarps/strains) can help to improve future biogeographic and ecological studies, as well as conservation strategies of species represented in the specimens deposited in herbaria and culture collections (Braga-Neto et al. 2013 Mycotaxon, 125(1), 289-301; Andrew et al. 2019 Phil. Trans. R. Soc. B, 374(1763), 20170392). These metadata information is even more important for new taxonomical description or for surveys of taxa in new geographical areas. Thus, I consider relevant to provide the geo-references (geographical coordinates) of the sampling sites where the described fungi were isolated.

2. In the line 65, I suggest changing the phrase “After testing purity…” for something like “Once obtaining an axenic culture of each isolate…”. I suggest the use of the “axenic culture” concept, which is more appropriate technical description for a fungal isolate growing alone in a culture.

Author Response

Dear Reviewer,

Dear Reviewer,

Thank you very much by the corrections/suggestions made to improve our manuscript.

At next, our responses.

Specific comments:

  1. Recent papers have emphasized the relevance to include georeferencing metadata associated to fungal taxonomic and biodiversity studies, georeferenced fungal specimens (sporocarps/strains) can help to improve future biogeographic and ecological studies, as well as conservation strategies of species represented in the specimens deposited in herbaria and culture collections (Braga-Neto et al. 2013 Mycotaxon, 125(1), 289-301; Andrew et al. 2019 Phil. Trans. R. Soc. B, 374(1763), 20170392). These metadata information is even more important for new taxonomical description or for surveys of taxa in new geographical areas. Thus, I consider relevant to provide the geo-references (geographical coordinates) of the sampling sites where the described fungi were isolated.

RESPONSE: Done

  1. In the line 65, I suggest changing the phrase “After testing purity…” for something like “Once obtaining an axenic culture of each isolate…”. I suggest the use of the “axenic culture” concept, which is more appropriate technical description for a fungal isolate growing alone in a culture.

RESPONSE: Done

Reviewer 2 Report

the reader may disagree with the use of ITS and perhaps other highly variable regions when comparing taxa across many genera (you even have different families!!!). unfortunately the reader has no chance to check the alignment as no alignment was deposited, e.g., in treebase and alignments were also not made available for review. the problem could be solved by using only conserved, well aligned loci (or conserved parts of the loci) for above species inferences. The species level phylogeny could be base on all characters, however, on the basis of clearly shorter datasets. 

the urgent recommendation is also to support the phylogenetic conclusions by doing phylogenetic analyses that are base on the individual loci. the individual phylogenetic trees could be shown as supplemental data

information related to phylogenetic distances are hardly informative for the reader and could be replaced by specifying numbers of nucleotide differences between closely related species. 

the reader might appreciate the information, which barcode or combination of such could best be used for using end point pcr strategy for identification purposes. 

there are a couple of strong statements that are not well supported by evidence. For example, the differently shaped phialides in Variabilis need to be also photographically documented.

additional comments: 

9-11: »probably because they have 9 not been considered a good subject of study due to, probably, an absence of discriminative morpho- 10 logical characters« does not seem to make sense or irrelevant.

11: Delete »Despite aero-aquatic fungi having been extensively studied,« knowledge --> Knowledge

14: terms »phoma-like and pyrenochaeta-like « not used in text. Why then in abstract?

34, plant debris. These are ubiquitous –> plant debris on ubiquitous

32, 34, 35: These, These, They… revise to make clear what you are talking about.

37-39: More or less the information was already provided in the previous sentences. Rewrite

40: Its (start of sentence). Rewrite so it is clear what you are talking about.

45: angustata, from –> angustata from, described the taxon Diplolaviopsis –> described Diplolaviopsis

71: A) after 14 d in the dark at 25±1 °C [17] –> A) incubated for 14 d in the dark at 25±1 °C [17]

72, 73: delete, was sort of written already in the previous sentence. Alternatively, rewrite.

80: Tokyo,Japan (space)

85: USA), with –> USA) with

90: unclear why tub2 is underlined

91: sentence ends without “.”

106: comparing of the –> comparing the

109-111: For tub2 sequences, a maximum level of identity (MLI) of < 98 % provides identification only at genus level, and a value > 98 % was considered to allow for species-level identification –> How do you know this?

114-116: rewrite to avoid “, respectively”

117: “the best nucleotide substitution model was General Time Reversible with Gamma ” –> this is most likely incorrect. You simply use this model in Raxml without knowing whether it is the best.

121: tub2 was the –> tub2 the

124: generations, four –> generations and four

139, 140: “The sequence datasets did not show conflict in the tree topologies for the 70 % reciprocal bootstrap trees”. If you even show the Blast results, consider also to show the single locus trees. It is hard to believe that ITS and the proteinencoding genes simply show the same topologies. Also, certain combinations of genes but not the combination of all may result into better phylogenetic resolutions.

143: after the burn-in with a stop value of 0.01 –> after removal of the burn-in and reaching a stop value of 0.01 (the latter needs to be better explained)

151: “all placed in independent terminal branches, representing two new species ”so you obtained only three strains from 100 samples (or )” –> being placed as independent terminal branches does not necessarily mean that you have new species. The intraspecific variation of sequences of closest sistertaxa, e.g., needs to be considered (here and elsewhere).

156: rewrite to avoid “respectively”

157: a new species each one –> a new species, of which each represented ….

167: remove “New”

173: 19. (1821). –> perhaps better: 19 (1821). (here and elsewhere)

185: make clear whether the described structures are from agar media or from natural substratum, especially because you have a separate chapter on culture characteristics (here and elsewhere).

188: 40-50 –> n-dash instead

190, 231, 286, etc.: Conidiomata wall 3–5-layered –> layered is unclear

193: replace “two morphologic types, ” with either and (194) “and” with “or”

198: are there not also conidia that are a bit longer than 3.5 microm?

211: “which are discrete and doliiform in N. acerina, N. aquatica and N. irregularis, ” How do you know? Perhaps the authors describing these species did not recognize that there were differently shaped conidiogenous cells.

214-216, 248-250, etc. (thus also elsewhere): better would be to also specify the number of differentiating nucleotides by comparing the species of the clade.

Fig. 2: the two kinds of conidiogenous cells are not convincingly illustrated but they have to be if you use the character for distinguishing different species.

For all coelomycetes described here: consider to describe also the “hymenium”

235: bacillar, slightly

244: unlike that N. variabilis –> unlike N. variabilis

244: the entire diagnosis needs to be rewritten. If the absence of dimorphic conidiogenous cells is important, the dimorphic characteristic of the cells in that other species needs to be better illustrated. Note, also other authors may follow your example! What about the numerous other species in the genus. Also colony characters, growth rates etc. should be used to better analyze the species differences. Which of the already described species have been studied and directly compared with the new species?

260: check correctness of the author list for the basionym. You use at least once “;”. Index Fungorum tells: “Neocucurbitaria prunicola Crous & Akulov 2019”. Accordingly, you may wish checking correctness of taxonomic authorities of all the other species.

275: neothailandica is unfortunate. Coin another eptithet.

300: than –> as

300: accordingly, it is fully unclear why neothailand. Is supposed to represent a new species.

Fig. 4: more than a single conidiogenous cell should be illustrated.

340: produces –> produce

341, 342: “(according those the observed in the 341 lectotype).” –> incomplete or corrupt sentences

Fig. 5 phialides are not sufficiently well illustrated. Hymenium could be better described.

374/5: lacking conidiomatous setae

396: immersed in what? Agar or naturals leaf substratum?

402: Conidiogenous cells / hymenium are/is insufficiently illustrated.

404: “3.5–5 :” according to the scale (Fig. 7g) conidia seem to be longer than 6 microm. Either scale or measurments might be wrong

416: “are covered with long setae in abundance[40] “–> are abundantly covered with long setae [40]

431-33: more evidence needs to be provided for the statement/claim

435: "well-developed conidiophores" check if this is correct, what kind of well-developed conidiophores (formed by coelomycetous fungi) would this be?

439: allowed them to recognize --> allowed recognition of

443: originally on --> ?originally described from

444: (a strain identified as belonging to 444 the species Pyrenochaeta acicola) --> unclear

457: the term "carbonaceous" was not used when describing the species and needs to be introduced/explained

461: but molecularly different --> but differing

472, 3 it seems that this sentence is wrongly written: fungi placing into

478: has been --> was

Author Response

Dear Reviewer,

Thank you very much by the corrections/suggestions made to improve our manuscript. 

At next, our responses.

Comments and Suggestions for Authors

the reader may disagree with the use of ITS and perhaps other highly variable regions when comparing taxa across many genera (you even have different families!!!). unfortunately the reader has no chance to check the alignment as no alignment was deposited, e.g., in treebase and alignments were also not made available for review. the problem could be solved by using only conserved, well aligned loci (or conserved parts of the loci) for above species inferences. The species level phylogeny could be base on all characters, however, on the basis of clearly shorter datasets.

RESPONSE: As noted by Jaklitsch et al. 2018 (https://doi.org/10.1016/j.simyco.2017.11.002), although established as the primary barcode of fungi for pragmatic reasons (Schoch et al. 2012), the ITS-region usually shows significantly less variation between closely related taxa in Cucurbitariaceae, Didymellaceae and their close families than protein-coding genes do, which provide a far better resolution in the case of closely related species. Consequently, we used four phylogenetically informative loci to build our phylogram, in the same way as other authors have done previously.

Wanasinghe D.N., Phookamsak R., Jeewon R., Wen Jing Li, Hyde K.D., Jones E.B.G., Camporesi E., Promputtha I. 2017. A family level rDNA based phylogeny of Cucurbitariaceae and Fenestellaceae with descriptions of new Fenestella species and Neocucurbitaria gen. nov. Mycosphere 8(4): 397–414. DOI: 10.5943/mycosphere/8/4/2

Valenzuela-Lopez N., Cano-Lira J.F., Guarro J., Sutton D.A., Wiederhold N., Crous P.W., Stchigel A.M. 2018. Coelomycetous Dothideomycetes with emphasis on the families Cucurbitariaceae and Didymellaceae. Studies in Mycology 90: 1–69. DOI: 10.1016/j.simyco.2017.11.003

Jaklitsch W.M., J. Checa, M.N. Blanco, I. Olariaga, S. Tello, H. Voglmayr. 2018. A preliminary account of the Cucurbitariaceae. Studies in Mycology 90: 71–118. DOI: 10.1016/j.simyco.2017.11.002

Jaklitsc, W.M., Voglmayr, H. 2020. Fenestelloid clades of the Cucurbitariaceae. Persoonia - Molecular Phylogeny and Evolution of Fungi 44: 1–40. DOI: 10.3767/persoonia.2020.44.01

However, we agree with the reviewer in that the alignment must be accessible to the researchers, which is why we uploaded the data to Treebase (http://purl.org/phylo/treebase/phylows/study/TB2:S28077), We have added this to the new version of the manuscript.

the urgent recommendation is also to support the phylogenetic conclusions by doing phylogenetic analyses that are based on the individual loci. the individual phylogenetic trees could be shown as supplemental data

RESPONSE: The placement of all known taxa into the Cucurbitariaceae, the Neopyrenochaetaceae and the Pyrenochaetopsiaceae was strongly supported, and the delimitation among the species was well-resolved in our concatenated tree. Only an individual tree (rpb2 sequence-based) displays a similar topology (but not the same degree of branch support). Despite that, the trees for each phylogenetically informative marker have now been included as supplemental material.

information related to phylogenetic distances are hardly informative for the reader and could be replaced by specifying numbers of nucleotide differences between closely related species.

RESPONSE: Done.

the reader might appreciate the information, which barcode or combination of such could best be used for using end point pcr strategy for identification purposes.

RESPONSE: We added the following to the Discussion: “Based on previous studies and in our experience with this group of fungi, we suggest using the ITS as the barcode for identification at genus level, and rpb2 for species recognition”.

there are a couple of strong statements that are not well supported by evidence. For example, the differently shaped phialides in Variabilis need to be also photographically documented.

RESPONSE: Two different coloured arrows have been added to Figure 2e in order to highlight these different shapes.

additional comments:

9-11: »probably because they have not been considered a good subject of study due to, probably, an absence of discriminative morphological characters« does not seem to make sense or irrelevant.

RESPONSE: Deleted.

11: Delete »Despite aero-aquatic fungi having been extensively studied,« knowledge --> Knowledge

RESPONSE: Done.

14: terms »phoma-like and pyrenochaeta-like « not used in text. Why then in abstract?

RESPONSE: Changed for “pycnidia-producing fungi”.

34, plant debris. These are ubiquitous –> plant debris on ubiquitous

RESPONSE: Because these fungi do not only grow on plant debris, we think it is not appropriate to do this.

32, 34, 35: These, These, They… revise to make clear what you are talking about.

RESPONSE: Modified.

37-39: More or less the information was already provided in the previous sentences. Rewrite

RESPONSE: Modified.

40: Its (start of sentence). Rewrite so it is clear what you are talking about.

RESPONSE: Modified.

45: angustata, from –> angustata from, described the taxon Diplolaviopsis –> described Diplolaviopsis

RESPONSE: Done.

71: A) after 14 d in the dark at 25±1 °C [17] –> A) incubated for 14 d in the dark at 25±1 °C [17]

RESPONSE: Done.

72, 73: delete, was sort of written already in the previous sentence. Alternatively, rewrite.

RESPONSE: Done.

80: Tokyo,Japan (space)

RESPONSE: Done.

85: USA), with –> USA) with

RESPONSE: Done.

90: unclear why tub2 is underlined

RESPONSE: Underline erased.

91: sentence ends without “.”

RESPONSE: Placed.

106: comparing of the –> comparing the

RESPONSE: Done.

109-111: For tub2 sequences, a maximum level of identity (MLI) of < 98 % provides identification only at genus level, and a value > 98 % was considered to allow for species-level identification –> How do you know this?

RESPONSE: These “breakpoints” have been proposed in previous articles (see above) to discriminate fungi at genus and at species level, respectively.

114-116: rewrite to avoid “, respectively”

RESPONSE: Done.

117: “the best nucleotide substitution model was General Time Reversible with Gamma ” –> this is most likely incorrect. You simply use this model in Raxml without knowing whether it is the best.

RESPONSE: Modified.

121: tub2 was the –> tub2 the

RESPONSE: Done.

124: generations, four –> generations and four

RESPONSE: Done.

139, 140: “The sequence datasets did not show conflict in the tree topologies for the 70 % reciprocal bootstrap trees”. If you even show the Blast results, consider also to show the single locus trees. It is hard to believe that ITS and the protein encoding genes simply show the same topologies. Also, certain combinations of genes but not the combination of all may result into better phylogenetic resolutions.

RESPONSE: Trees for each phylogenetically informative marker are now available as supplementary material. The reviewer is right to suppose that the topology of the ITS-based tree does not match those based on gene-encoding proteins, mostly because not all the species included in our study have nucleotide sequences for all genes. Consequently, we have considered appropriate to use all the available information in a concatenated tree, as it was used in previous studies to increase the reliability of the results obtained.

143: after the burn-in with a stop value of 0.01 –> after removal of the burn-in and reaching a stop value of 0.01 (the latter needs to be better explained)

RESPONSE: Done

151: “all placed in independent terminal branches, representing two new species” so you obtained only three strains from 100 samples (or )” –> being placed as independent terminal branches does not necessarily mean that you have new species. The intraspecific variation of sequences of closest sister taxa, e.g., needs to be considered (here and elsewhere).

RESPONSE: The mention of which “independent terminal branches” represent “new species” is deleted from the new version of the manuscript. However, the authors want to explain that the intraspecific variation (e.g. Cucurbitaria berberidis, and Neopyrenochaeta annellidica and N. maesuayenis) of the nucleotide sequences of four different genetic markers is very rare in the Cucurbitariaceae and the Neopyrenochaetaceae families.

156: rewrite to avoid “respectively”

RESPONSE: Done.

157: a new species each one –> a new species, of which each represented ….

RESPONSE: See response to line note 151.

167: remove “New”

RESPONSE: Done.

173: 19. (1821). –> perhaps better: 19 (1821). (here and elsewhere)

RESPONSE: Done.

185: make clear whether the described structures are from agar media or from natural substratum, especially because you have a separate chapter on culture characteristics (here and elsewhere).

RESPONSE: Modified. See lines 74 and 75 of the new version of the manuscript.

188: 40-50 –> n-dash instead

RESPONSE: Done.

190, 231, 286, etc.: Conidiomata wall 3–5-layered –> layered is unclear

RESPONSE: Modified.

193: replace “two morphologic types, ” with either and (194) “and” with “or”

RESPONSE: Done.

198: are there not also conidia that are a bit longer than 3.5 microm?

RESPONSE: No, there is not.

211: “which are discrete and doliiform in N. acerina, N. aquatica and N. irregularis, ” How do you know? Perhaps the authors describing these species did not recognize that there were differently shaped conidiogenous cells.

RESPONSE: Because we are sure that the authors who described N. acerina, N. aquatica and N. irregularis are competent taxonomists, and the presence of more than one conidiogenous cell shape is an important “classic” taxonomic criterion in this group of fungi, we assume that the description of a unique morphology for the conidiogenous cell is reliable.

214-216, 248-250, etc. (thus also elsewhere): better would be to also specify the number of differentiating nucleotides by comparing the species of the clade.

RESPONSE: Done.

Fig. 2: the two kinds of conidiogenous cells are not convincingly illustrated but they have to be if you use the character for distinguishing different species.

RESPONSE: In the new version of the manuscript different-coloured arrows show the two sorts of phialides.

For all coelomycetes described here: consider to describe also the “hymenium”

RESPONSE: Because the “hymenium” in these sorts of coelomycetous fungi are only made up of conidiogenous cells/conidiophores and supporting cells, we consider our description complete.

235: bacillar, slightly

RESPONSE: Done.

244: unlike that N. variabilis –> unlike N. variabilis

RESPONSE: Done.

244: the entire diagnosis needs to be rewritten. If the absence of dimorphic conidiogenous cells is important, the dimorphic characteristic of the cells in that other species needs to be better illustrated. Note, also other authors may follow your example! What about the numerous other species in the genus. Also colony characters, growth rates etc. should be used to better analyze the species differences. Which of the already described species have been studied and directly compared with the new species?

RESPONSE: The diagnosis has been re-written. In the new version of the manuscript Fig. 2e has two arrows that show as clearly as possible both morphological types of phialides.

260: check correctness of the author list for the basionym. You use at least once “;”. Index Fungorum tells: “Neocucurbitaria prunicola Crous & Akulov 2019”. Accordingly, you may wish checking correctness of taxonomic authorities of all the other species.

RESPONSE: Revised and modified throughout the manuscript.

275: neothailandica is unfortunate. Coin another eptithet.

RESPONSE: Changed by Neopyrenochaeta asexualis.

300: than –> as

RESPONSE: Modified.

300: accordingly, it is fully unclear why neothailandica. Is supposed to represent a new species.

RESPONSE: Substituted by the epithet “asexualis”.

Fig. 4: more than a single conidiogenous cell should be illustrated.

RESPONSE: Done.

340: produces –> produce

RESPONSE: Modified.

341, 342: “(according those the observed in the 341 lectotype).” –> incomplete or corrupt sentences

RESPONSE: Modified.

Fig. 5 phialides are not sufficiently well illustrated. Hymenium could be better described.

RESPONSE: For better visualization, we have added an arrow to show the phialide location. The description of “hymenium” has been clarified.

374/5: lacking conidiomatous setae

RESPONSE: Modified.

396: immersed in what? Agar or naturals leaf substratum?

RESPONSE: Immersed in agar. M&M now states that vegetative and reproductive structures were described on oatmeal agar.

402: Conidiogenous cells / hymenium are/is insufficiently illustrated.

RESPONSE: Fig. 7e and 7f were replaced.

404: “3.5–5 :” according to the scale (Fig. 7g) conidia seem to be longer than 6 microm. Either scale or measurments might be wrong

RESPONSE: Modified.

416: “are covered with long setae in abundance[40] “–> are abundantly covered with long setae [40]

RESPONSE: Modified.

431-33: more evidence needs to be provided for the statement/claim

RESPONSE: Modified.

435: "well-developed conidiophores" check if this is correct, what kind of well-developed conidiophores (formed by coelomycetous fungi) would this be?

RESPONSE: The presence of conidiophores is a recognized fact in certain coelomycetous fungi, e.g. Neocucurbitaria (formerly Pyrenochaeta) unguis-hominis (https://core.ac.uk/download/pdf/29240399.pdf).

439: allowed them to recognize --> allowed recognition of

RESPONSE: Modified.

443: originally on --> ?originally described from

RESPONSE: Modified.

444: (a strain identified as belonging to the species Pyrenochaeta acicola) --> unclear

RESPONSE: Modified.

457: the term "carbonaceous" was not used when describing the species and needs to be introduced/explained

RESPONSE: Modified.

461: but molecularly different --> but differing

RESPONSE: Modified.

472, 3 it seems that this sentence is wrongly written: fungi placing into

RESPONSE: Modified.

478: has been --> was

RESPONSE: Modified.